# Association Between Activated Loci of HML-2 Primate-Specific Endogenous Retrovirus and Newly Formed Chromatin Contacts in Human Primordial Germ Cell-like Cells

**DOI:** 10.3390/ijms252413639

**Published:** 2024-12-20

**Authors:** Bianca Cordazzo Vargas, Toshihiro Shioda

**Affiliations:** 1Krantz Family Center for Cancer Research, Massachusetts General Hospital, Charlestown, MA 02114, USA; 2Computational Biomedicine, New York University Grossman School of Medicine, New York, NY 10016, USA

**Keywords:** PGCLC, HERV-K, topologically associated domains, Hi-C

## Abstract

The pluripotent stem cell (PSC)-derived human primordial germ cell-like cells (PGCLCs) are a cell culture-derived surrogate model of embryonic primordial germ cells. Upon differentiation of PSCs to PGCLCs, multiple loci of HML-2, the hominoid-specific human endogenous retrovirus (HERV), are strongly activated, which is necessary for PSC differentiation to PGCLCs. In PSCs, strongly activated loci of HERV-H family HERVs create chromatin contacts, which are required for the pluripotency. Chromatin contacts in the genome of human PSCs and PGCLCs were determined by Hi-C sequencing, and their locations were compared with those of HML-2 loci strongly activated in PGCLCs but silenced in the precursor naïve iPSCs. In both iPSCs and PGCLCs, the size of chromatin contacts were found to be around one megabase, which corresponds to the Topologically Associated Domains in the human genome but is slightly larger in PGCLCs than iPSCs. The number of small-sized chromatin contacts diminished while numbers of larger-sized contacts increased. The distances between chromatin contacts newly formed in PGCLCs and the degrees of activation of the closest HML-2 loci showed significant inverse correlation. Our study provides evidence that strong activation of HML-2 provirus loci may be associated with newly formed chromatin contacts in their vicinity, potentially contributing to PSC differentiation to the germ cell lineage.

## 1. Introduction

Human primordial germ cells (PGCs) emerge from the embryonic epiblast and amnion around implantation as the earliest precursors of all germ cells [1,2,3,4,5]. As access to PGCs in human embryos is extremely challenging technically and ethically, the use of the pluripotent stem cell (PSC)-derived PGC-like cells (PGCLCs) as a PGC surrogate model is becoming increasingly popular for in vitro investigation of the biological characteristics of PGCs [6,7]. The conventional human PGCLCs resemble early-stage PGCs whose genomic DNA still retains cytosine methylations at levels comparable to their precursors, whereas recent studies have produced more advanced stages of PGCLCs whose global DNA methylation is strongly diminished, resembling the massive epigenetic reprogramming characteristics of the embryonic PGCs [8,9].

In the tightly packed nucleus of human cells, interphase chromosomes occupy distinct territories to create a three-dimensional genome organization that supports higher-order chromatin interactions, which strongly affects the coordinated expression of genes [10]. Two strongly interacting regions on the same chromosome, around one megabase apart, can create the topologically associated domains (TADs), which are structurally and functionally isolated regions strongly conserved across different types of cells in the same organism [11,12,13]. However, a recent study revealed that, in human PSCs, strong transcriptional activity of several loci of the human endogenous retrovirus (HERV)-H family (HERV-H) creates TADs that are required for the maintenance of pluripotency [14]. Because differentiation of human PSCs to PGCLCs involves strong activation of the HML-2 hominoid-specific HERV species [15,16,17], our interest lies in whether the activated HML-2 can create TADs as well.

In the current study, we identified TADs in the genomes of human PGCLCs and their precursor iPSCs by Hi-C sequencing and compared their sizes and numbers. Locations of the TAD boundaries newly formed upon iPSC differentiation to PGCLCs were compared with locations of known HML-2 loci that are silenced in iPSCs but activated in PGCLCs. The outcomes of our analysis show that most strongly activated HML-2 loci are associated with newly formed TAD boundaries, supporting our hypothesis that HML-2 activation may alter the genomic landscape of TADs in the PGC(LC)s.

## 2. Results

The workflow of the present study is shown in Figure 1A. Human PGCLC cultures were generated from naïve iPSC clones in three independent experiments. Genomic DNA of PGCLCs and iPSCs were subjected to Hi-C sequencing to generate contact matrices. TADs were mapped on the GRCh38/hg38 human genome reference sequence using ClusterTAD, which uses an unsupervised machine-learning algorithm to determine the statistically most credible locations of TAD boundaries [18].

Using the best-estimated TADs identified by ClusterTAD, we compared the distribution of TAD sizes in PGCLCs and iPSCs, as well as the frequency of TADs that fall within specific size intervals between these cell types. The size of TADs in the human genome is estimated to be about 1 megabase [13]. Consistent with this prior estimation, the sizes of TADs estimated in our analysis were around 1 megabase and largely comparable between PGCLCs and iPSCs (Figure 1B), which was expected, as TADs are typically conserved beyond cell types [10,11,12,13]. However, we observed a significant increase in TAD sizes in PGCLCs compared to iPSCs in chromosome 3 (*p* < 0.05), chromosome 4 (*p* < 0.05), chromosome 8 (*p* < 0.01), and chromosome 17 (*p* < 0.05). Detailed analysis on the sizes of chromatin contacts also detected significant increases in TAD frequencies in PGCLCs in groups with relatively large chromatin contacts (*p* < 0.05 for 1.5–2 Mb, and *p* < 0.001 for 2–2.5 Mb), while conversely, TAD frequencies at relatively small chromatin contacts (200 kb–1 Mb size group) decreased significantly (*p* < 1 × 10^−^⁴) (Figure 1C). These results suggest that the genome-wide number of the chromatin contact boundaries decreased upon iPSC differentiation to PGCLCs.

We next identified differential chromatin interactions (DCIs) newly created upon differentiation of iPSCs to PGCLCs using the BART3D software tool, which is specifically designed to estimate DCIs between two Hi-C matrices. As shown in Figure 2A, the BART3D-identified DCIs were mostly consistent across the three biological experimental replicates. Then locations of the consensus DCIs (second innermost ring of Figure 2A) were compared with the locations of the top 19 most strongly expressed HML-2 loci in PGCLCs (innermost ring) [15]. We observed a limited degree of overlap between the consensus DCIs and the active HML-2 loci. We further selected the consensus DCIs estimated by BART3D for their close overlap with the TAD boundaries estimated in clusterTAD (Figure 2B). The selected DCIs were strongly conserved across the three biological replicates of PGCLC cultures; however, their overlap with PGCLCs was still limited.

In an attempt to obtain a further link between the selected DCIs and the HML-2 loci activated in PGCLCs, we examined whether the strength of HML-2 activation can affect the distance of HML-2 and the closest DCI (Figure 2C). Strikingly, the three most strongly expressed HML-2 loci were located within close proximities of DCIs (Expression Rank 1, large blue dots), and the X-Y plot between the strength of HML-2 activation (Y-axis) and the distance to the closest DCI (X-axis) showed strong inverse correlation (Spearman coefficient: −0.5953, *p* = 0.0072), suggesting that stronger HML-2 activation may cause creation of a new chromatin contact at a closer distance. This trend was confirmed by a permutation test (Figure 2D), which demonstrates that the distance between strongly expressed HML-2 loci (Expression Ranks 1 and 2 in Figure 2C) and their closest DCIs are very significantly smaller than expected from simulated random distributions (*p* = 0.003), whereas inclusion of less strongly expressed HML-2 loci (Expression Ranks 1–4) lost the statistical significance (*p* = 0.12).

## 3. Discussion

Our current study provided an insight that (1) iPSC differentiation to PGCLCs is associated with a small degree of changes in chromatin contact landscape, and (2) strong activation of HML-2 in PGCLCs may contribute creation of new chromatin contacts in their vicinities. We identified ~120 DCIs and ~80 differential TADs between human PGCLCs and their precursor iPSCs (Figure 2A,B). Among the top 19 strongly activated HML-2 provirus loci identified in our previous study [15], the three most strongly expressed loci were located in a statistically significant close vicinity of the differential TAD boundaries (1–7 megabases; Expression Rank 1 in Figure 2C). More weakly expressed HML-2 provirus loci tend to be more distant from the differential TAD boundaries (Figure 2C). The association between the strength of RNA expression from the HML-2 provirus loci and the newly formed TAD boundaries was also statistically significant: the Spearman’s correlation test (*p* = 0.0072) and a permutation test (*p* = 0.003) are shown in Figure 2C,D, respectively. Based on these observations, we speculate that strong activation of HML-2 proviruses may contribute to the creation of new TAD boundaries in PGCLCs. However, because the observed associations are relatively low, our current study could not determine whether HML-2 activation directly contributes to the formation of new TAD boundaries. It is possible that strong activation of HML-2 opens otherwise tightly closed chromatin, thus increasing the chance of TAD formation in the relatively close vicinity but via different mechanisms that may not be directly tied with the HML proviruses. On the other hand, it is also possible that formation of new TAD boundaries via different mechanisms reduces silencing of HML-2 proviruses by heterochromatin. Future studies will be necessary to clarify the mechanisms behind the apparent association between HML-2 provirus activation and the formation of new TAD boundaries. The number of TAD boundaries whose creation may be associated with the activated loci of HML-2 proviruses is only a fraction of the number of all TAD boundaries or DCIs. In our preceding study, we identified activated HML-2 provirus loci using RNA-seq followed by ERVmap2 locus-specific quantitation of viral RNA [15]. This approach was unable to detect potentially activated solo-LTR loci of HML-2 due to the lack of viral RNA expression. Because the number of solo-LTR HML-2 loci (~1000) is far larger than that of RNA-expressing HML-2 proviruses (~100) [19], the differential TADs or DCIs that were not linked to HML-2 provirus activation in the current study could still be associated with the activation of solo-LTR HML-2 loci. Future study will be necessary to determine the complete links between the activated loci of HML-2 proviruses and solo-LTRs and the formation of TADs or DCIs upon differentiation of human PSCs to PGC(LC)s.

Zhang et al. reported that strongly activated loci of HERV-H in human PSCs are necessary for the cellular pluripotency through the creation of new TAD boundaries, which affects the expression of pluripotency-associated genes [14]. They presumed that other types of HERV loci may also be capable of creating new chromatin contacts as long as they are transcriptionally activated very strongly. Our data are consistent with their prediction, potentially contributing to possible generalization of the notion that strongly activated HERVs can create chromatin contact in their vicinities. However, in our current study, analyses on distances between DCIs and HERV-H loci were challenging due to far larger numbers of differentially expressed HERV-H loci than HML-2, resulting in greater degrees of inter-experimental variability in our three biological replicates. Our similar analyses focusing on DCIs lost upon differentiation of iPSCs to PGCLCs did not reveal any significant associations between the DCI locations and activated HML-2, either. Further studies are necessary to extend the scope of this notion, as well as understand the detailed mechanisms behind it.

The expression of HERVs is typically suppressed in healthy human cells except for PSCs and PGCLCs. Human embryonic stem cells (ESCs) and induced pluripotent stem cells (iPSCs) strongly express HERV-H [14], and human PGCLCs express HML-2 [15,16,17]. Our recent study examined RNA expression from all known HERV species in human iPSCs and PGCLCs and observed exclusive expression of HERV-H and HML-2 [15]. Thus, the expression of HERV-H was stronger in human iPSCs but still significant in PGCLCs. In contrast, the expression of HML-2 was limited to PGCLCs. Thus, among all HERV families, HERV-H and HML-2 seem to have most significant roles, if any, in PGCLC differentiation.

The functional importance of HML-2 activation in human PGC specification was previously suggested by Xiang et al. [16], who suppressed the transcriptional activity of HML-2 in the genome of human embryonic stem cells by CRISPRi. In their study, HML-2 transcription was suppressed by a lentivirus-based CRISPRi system involving a dCas9-fusion KRAB transcriptional repressor and two gRNA species targeting a consensus sequence of HML-2. The CRISPRi suppression of HML-2 resulted in a significantly reduced efficiency of PGCLC induction (15.4% to 4.1%), demonstrating the importance of HML-2 activation in the PGCLC differentiation of human PSCs. The authors predicted that some HML-2 loci in the human genome may function as the TEENhancers (Transposable Element-Embedded eNhancers) to activate the genes necessary for germline specification of human PSCs, although a direct link between locus-specific HML-2 activation and PGCLC specification through the TEENhancer activities remains to be demonstrated. Our current study provides an alternative mechanism by which HML-2 activation may contribute to PGCLC specification—namely, highly activated HML-2 can remodel the 3-D genome structure, which may affect the expression of the germline specification genes. The 3-D genome remodeling and the TEENhancer activities are not mutually exclusive and may occur simultaneously. Future studies are necessary to identify the HML-2 loci contributing to the germline specification and their mode of actions.

Three distinct types of Long Terminal Repeats (LTRs) are found in the HML-2 loci—namely, LTR5_Hs, LTR5A, and LTR5B [19,20]. LTR5_Hs, the youngest HERV species, entered into the genome of hominoids (i.e., apes, including humans, chimpanzees, gorillas, orangutans, and gibbons) after they had diverted from the old-world monkeys (OWMs). Thereafter, LTR5_Hs was amplified in the hominoid genome even after the human–chimpanzee divergence. Ito et al. performed comparative analysis between human PGCLCs and in vivo PGCs of the crab-eating macaque, a representative OWM lacking LTR5_Hs [17]. Because several genes upregulated in human PGCLCs but not in macaque PGCs located in the close proximity of LTR5_Hs loci activated in human PGCLCs, they presumed that such genes, which were significantly enriched to glucose metabolism and oxidative phosphorylation, may be regulated by the associating active LTR5_Hs loci. The authors further speculated that LTR5_Hs insertions may have altered expression of their adjacent genes towards the pattern specific to the hominoid PGCs/PGCLCs, playing a unique evolutionary role in hominoid PGC development. Interestingly, our recent study revealed that HML-2 loci activated in human PGCLCs harbored LTR5_Hs in both 5′ and 3′ LTRs, whereas HML-2 containing LTR5A or LTR5B at one or two viral LTRs remained silent [15], suggesting that the HML-2/LTR5_Hs loci may have unique significance in the evolutional aspects of PGC development, whereas contributions of the older HML-2 loci common to hominoids and OWM seem limited.

Our current study provides evidence that strongly activated HML-2 loci in the genome of human PGCLCs may cause local creation of chromatin contacts, laying an additional layer of epigenomic mechanism that can contribute to the lineage specification of PSCs towards PGC(LC)s. Contributions of HML-2 in PGC(LC) differentiation may also provide insights into the evolutionary perspective of germline lineage specification in primates.

## 4. Materials and Methods

Cell cultures—A flow chart of the experimental steps is presented as Figure 1A. Human PGCLCs were generated from the human iPSC clones A4, A5, and A6 using the protocol we previously described [6,7,21,22]. These three iPSC clones and the human PGCLCs derived from them were identical to the cells examined in our preceding study on the expression of viral RNA from HERV loci [15]. The human iPSCs were all males and generated from fibroblasts using a footprint-free method involving episomally replicating plasmid expression vectors that do not integrate into the genomic DNA and are spontaneously removed from completely reprogrammed cells (Human iPS Cell Reprogramming Episomal Kit, Cat# RF202, ALSTEM, Richmond, CA, USA) [7]. Normal diploid genomes and the competence of pluripotent differentiation of the human iPSCs were confirmed by Whole Genome Sequencing and a teratoma formation assay, respectively. Biological replicate PGCLC cultures were generated in three independent experiments performed at different times.

Hi-C sequencing—Genomic DNA of PGCLCs and the naïve pluripotency iPSCs were subjected to proximal ligation with formaldehyde followed by Hi-C deep sequencing library production [23] using the Arima Hi-C Kit (Cat# A510008, Arima Genomics, Carlsbad, CA, USA). The libraries were sequenced using Illumina NovaSeq6000 sequencer (Illumina, San Diego, CA, USA) for 150 nt + 150 nt paired-end reads.

Hi-C data analysis—The Hi-C FASTQ data were quality-filtered using the Trim Galore (version 0.6.0) and FastQC (version 0.11.9) software tools distributed by The Babraham Institute of Cambridge University (Cambridge, UK). The high-quality FASTQ data (Phred score ≥ 20, Illumina adaptor sequences removed) were subjected to contact matrix generation using the Hi-C Pro software (version 2.11.1) with the default setting, including the ICE normalization [24]. From the normalized contact matrix data, TADs were mapped on the human genome reference sequence (GRCh38/hg38) using ClusterTAD (version 1.0) [18]. The BART3D tool (version 1.1) [25] was used to identify differential chromatin interactions (DCIs). DCIs conserved across the three pairs of biological replicates (PGCLC vs. iPSC) were identified, and their distances to the nearest TAD boundaries estimated by ClusterTAD were evaluated using in-house computer scripts written in Python (version 3.6.10). The DCIs supported by both BART3D and ClusterTAD analyses of the three biological replicates were subjected to estimation of their distance from activated HML-2 loci located in the closest vicinities (locations of the active HML-2 loci are presented in our separate study [15]). Circos plots were generated using Circa software (OMGenomics Labs, San Francisco, CA, USA, version 1.2.1). Scatter plots and histograms were drawn using R.

Statistics—Statistical significance between the two groups was established using the Mann–Whitney U test via the statannotations package (version 0.6.0). Correlation between RNA expression from each HML-2 loci and their distance from the nearest TAD boundary was evaluated by calculating Spearman’s correlation coefficient using scipy (version 1.5.2), and the statistical significance of the correlation was tested using Spearman’s rank–order correlation test. The significance of the hypothesis H1—The distances between the strongly expressed HML-2 provirus loci and their nearest TAD boundaries are shorter than expected from random distributions of HML-2 proviruses in the human genome—was evaluated by a permutation test that calculated distances from randomly assigned locations in the human genome to the nearest TAD boundaries 2000 times.

## 5. Conclusions

We identified chromatin contacts in the genomes of human naïve PSCs and PGCLCs by Hi-C sequencing. Upon differentiation of PSCs to PGCLCs, the genome-wide number of TADs was slightly decreased while the size of TADs increased in a few chromosomes, although the overall 3-D genome structure was largely maintained. Strongly activated HML-2 provirus loci tended to be located in the vicinity of newly created TAD boundaries, suggesting possible roles of HML-2 activation in the formation of chromatin contacts.

## Figures and Tables

**Figure 1 ijms-25-13639-f001:**
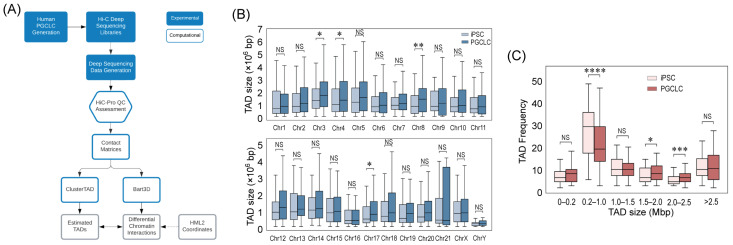
Global changes in TAD size upon differentiation of human naïve iPSCs to PGCLCs. (**A**) Flowchart of data acquisition (filled steps) and analysis (open steps). (**B**) Distribution of TADs across chromosomes. (**C**) Genome-wide distribution of TAD frequencies in discrete size clusters. For (**B**,**C**), Mann–Whitney U test outcomes are reported (* for 0.01 < *p* ≤ 0.05; ** for 0.001 < *p* ≤ 0.01; *** for 0.0001 < *p* ≤ 0.001; **** for *p* ≤ 0.00001). NS, not significant.

**Figure 2 ijms-25-13639-f002:**
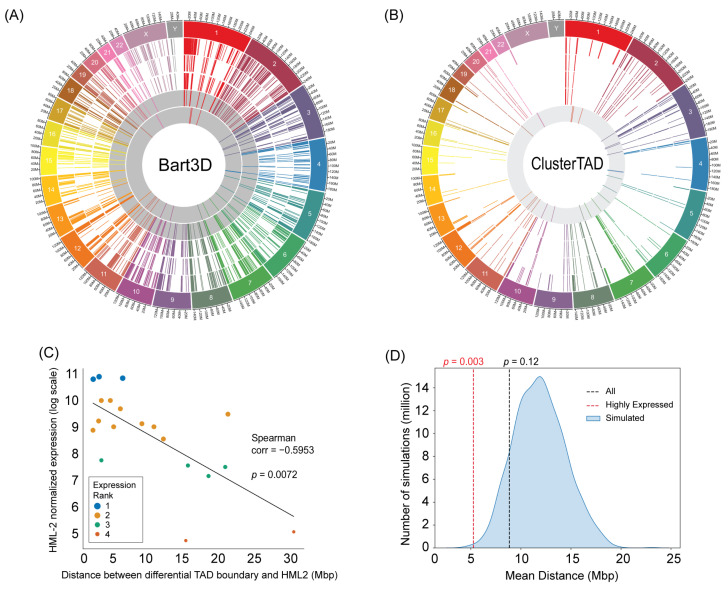
(**A**,**B**) Circos plots illustrating the locations of conserved genomic regions and HML-2 sequences. (**A**) Conserved DCIs identified via BART3D across biological replicates. The three outermost rings depict DCIs with increased interactions in PGCLCs compared to naive iPSCs (A4, A5, and A6 clones from the inner to outer rings). The fourth ring (gray background) highlights DCIs that are conserved across the previous three rings. The innermost ring (gray background) shows HML-2 locations. (**B**) Best TADs estimated with ClusterTAD that are closest to conserved DCIs. The outermost ring displays the conserved DCIs across replicate pairs. The next three rings show the best TADs from the three PGCLC biological replicates nearest to these conserved DCIs (A4, A5, and A6 clones from the inner to outer rings). The innermost ring (gray background) shows HML-2 locations. DCIs: differential chromatin interactions. (**C**) Correlation between the proximity (in base pairs) of highly conserved regions and HML-2 sequences with HML-2 normalized counts (Spearman ρ = −0.5933, *p* < 0.0072). Only HML-2 sequences paired with a unique conserved region are shown (*n* = 19), with dot sizes proportional to their expression ranks (1 being the most highly expressed and 4 the lowest). Counts are log-scaled for improved visualization. (**D**) Permutation test for the distance between HML-2 regions (*n* = 19) and conserved genomic regions. The density plot illustrates the distribution of distances under the null hypothesis, where conserved genomic regions (*n* = 19) were randomly scattered across the genome. The black dashed line represents the observed mean distances between HML-2 sequences and conserved genomic regions with increased chromatin interactions in PGCLCs. The red dashed lines represent the same metrics calculated using only HML-2 sequences with an expression rank of 1 or 2.

## Data Availability

Hi-C sequencing data are available from the NCBI GEO database (accession number: GSE284598). Computational scripts are available in the following GitHub repository: https://github.com/Biacordazzo/hPGCLC_HiC, accessed on 10 December 2024.

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
