# Peer review of "Association Between Activated Loci of HML-2 Primate-Specific Endogenous Retrovirus and Newly Formed Chromatin Contacts in Human Primordial Germ Cell-like Cells"

_ijms, 2024, doi:10.3390/ijms252413639_

Round 1
Reviewer 1 Report
Comments and Suggestions for Authors
This study investigates the role of HML-2, a hominoid-specific human endogenous retrovirus (HERV), in the differentiation of human primordial germ cell-like cells (PGCLCs) from induced pluripotent stem cells (iPSCs). Using Hi-C sequencing, the authors analyzed chromatin contact patterns in iPSCs and PGCLCs, examining changes in topologically associated domains (TADs). The findings suggest that HML-2 activation correlates with newly formed chromatin contacts, potentially influencing PGCLC differentiation. However, substantial improvement is needed.
Major:
1. Although this study found potential interesting observations, functional assays to directly link HML-2 activation with outcomes of PGCLCs induction are needed.
2. While HML-2 is the primary focus, examining other HERV families might provide a broader perspective on retroviral influence in PGCLC differentiation.
Minor:
1. Although the study focuses on human-specific retroviruses, insights from other primates could contextualize HML-2's unique role and evolutionary significance in human PGCLC development.
Author Response
Major Comment 1: Although this study found potential interesting observations, functional assays to directly link HML-2 activation with outcomes of PGCLCs induction are needed.
Response to Major Comment 1: In a previously published study, which was cited in the original manuscript as Ref# 16, Xiang et al. suppressed the transcriptional activity of HML-2 loci in the genome of human embryonic stem cells by CRISPRi and observed a significantly reduced efficiency of PGCLC induction. The outcome of this experiment supports the functional importance of HML-2 activation in PGCLC induction from human pluripotent stem cells. However, the detailed mechanisms through which HML-2 activation contributes to PGCLC induction still remain to be elucidated although Xiang et al. speculated the role of HML-2 as the TEENhancers (Transposable Element-Embedded eNhancers) for activation of germline specification genes.
Our present study suggests an alternative mechanism by which HML-2 activation may contribute to human pluripotent stem cell differentiation to germline lineage through remodeling of the chromatin contacts. Although we admit that our current manuscript does not provide data directly linking such 3-D genome structure remodeling and the observed HML-2 activation, additional experiments to obtain such data will be beyond a typically restricted scope of studies presented in the “Brief Report” publication type. The very short period of time that the journal Editor allowed us for preparation of a revision (10 days) did not allow us to obtain such new data, either. Under the current circumstance, to address the review comment we added the following paragraph in Discussion of the revised manuscript.
The functional importance of HML-2 activation in human PGC specification was previously suggested by Xiang et al. [16], who suppressed the transcriptional activity of HML-2 in the genome of human embryonic stem cells by CRISPRi. In their study, HML-2 transcription was suppressed by a lentivirus-based CRISPRi system involving a dCas9-fusion KRAB transcriptional repressor and two gRNA species targeting a consensus sequence of HML-2. The CRISPRi suppression of HML-2 resulted in a significantly reduced efficiency of PGCLC induction (15.4% to 4.1%), demonstrating the importance of HML-2 activation in PGCLC differentiation of human PSCs. The authors predicted that some HML-2 loci in the human genome may function as the TEENhancers (Transposable Element-Embedded eNhancers) to activate genes necessary for germline specification of human PSCs although direct link of locus-specific HML-2 activation and PGCLC specification through the TEENhancer activities remains to be demonstrated. Our current study provides an alternative mechanism by which HML-2 activation may contribute to PGCLC specification – namely, highly activated HML-2 can remodel the 3-D genome structure, which may affect expression of the germline specification genes. The 3-D genome remodeling and the TEENhancer activities are not mutually exclusive and may occur simultaneously. Future studies are necessary to identify the HML-2 loci contributing to the germline specification and their mode of actions.
Comment 2: While HML-2 is the primary focus, examining other HERV families might provide a broader perspective on retroviral influence in PGCLC differentiation.
Response to Major Comment 2: Our recent study, which is cited in our original manuscript as Ref# 15, examined expression of viral RNA from all known species of HERVs in human iPSCs and PGCLCs. The study found exceptionally strong RNA expression from only HML-2, which belongs to the HERV-K family, and the HERV-H family. Expression of HERV-H was stronger in iPSCs although it was still significant in PGCLCs. In contrast, expression of HML-2 was exclusively observed with PGCLCs. Thus, among all HERV families, HERV-H and HML-2 seem to have most significant roles, if any, in PGCLC differentiation. To address the review comment, we added the following paragraph in Discussion of the revised manuscript.
Expression of HERVs is typically suppressed in healthy human cells except for PSCs and PGCLCs. Human embryonic stem cells (ESCs) and induced pluripotent stem cells (iPSCs) strongly express HERV-H [14], and human PGCLCs express HML-2 [15-17]. Our recent study examined RNA expression from all known HERV species in human iPSCs and PGCLCs and observed exclusive expression of HERV-H and HML-2 [15]. Thus, expression of HERV-H was stronger in human iPSCs but still significant in PGCLCs. In contrast, expression of HML-2 was limited to PGCLCs. Thus, among all HERV families, HERV-H and HML-2 seem to have most significant roles, if any, in PGCLC differentiation.
Minor Comment 1: Although the study focuses on human-specific retroviruses, insights from other primates could contextualize HML-2's unique role and evolutionary significance in human PGCLC development.
Response to Minor Comment 1: The full-length sequence of the HML-2 human endogenous retrovirus consists of a protein-coding region flanked by a 5’-LTR (Long Terminal Repeat) and a 3’-LTR. Three distinct LTRs are found in the HML-2 loci – namely, LTR5_Hs, LTR5A, and LTR5B. LTR5_Hs, the youngest human endogenous retrovirus subfamily, expanded in the hominoid lineage (i.e., apes, including humans, chimpanzees, gorillas, orangutans, and gibbons) but not in the old world monkeys (OWMs), which diverted from the hominoids before LTR5_Hs entered into the hominoid genome. LTR5_Hs amplified in the hominoid genome even after the human-chimpanzee divergence.
In a previous study cited in our original manuscript as Ref# 17, Ito et al. performed comparative analysis between human PGCLCs and in vivo PGCs of the crab-eating macaque, a representative OWM lacking LTR5_Hs. Because several genes upregulated in human PGCLCs but not in macaque PGCs located in the close proximity of LTR5_Hs activated in human PGCLCs, the authors presumed that such genes, which were significantly enriched to glucose metabolism and oxidative phosphorylation, may be regulated by the associating loci of LTR5_Hs. The authors further speculated that LTR5_Hs insertions may have altered expression of their adjacent genes towards the pattern specific to the hominoid PGCs/PGCLCs, playing a unique evolutionary role in hominoid PGC development.
Interestingly, our own recent study cited in our manuscript as Ref# 15 revealed that HML-2 loci activated in human PGCLCs harbored LTR5_Hs in both 5’ and 3’ LTRs whereas HML-2 containing LTR5A or LTR5B at one or two viral LTRs remained silent. This observation further supports the notion that the HML-2/LTR5_Hs loci that entered the hominoid genome after the hominoid-OWM divergence may have unique significance in the evolutional aspects of PGC development whereas contributions of the older HML-2 loci common to hominoids and OWM seem limited.
Responding to the review comment, we added the following new paragraph in Discussion.
Three distinct types of Long Terminal Repeats (LTRs) are found in the HML-2 loci – namely, LTR5_Hs, LTR5A, and LTR5B [new reference: Xue et al. 2020 Retrovirology 17:10, and Subramanian et al. 2011 Retrovirology 8:90]. LTR5_Hs, the youngest HERV species, entered in the genome of hominoids (i.e., apes, including humans, chimpanzees, gorillas, orangutans, and gibbons) after they had diverted from the old world monkeys (OWMs). Thereafter, LTR5_Hs amplified in the hominoid genome even after the human-chimpanzee divergence. Ito et al. performed comparative analysis between human PGCLCs and in vivo PGCs of the crab-eating macaque, a representative OWM lacking LTR5_Hs [17]. Because several genes upregulated in human PGCLCs but not in macaque PGCs located in the close proximity of LTR5_Hs loci activated in human PGCLCs, they presumed that such genes, which were significantly enriched to glucose metabolism and oxidative phosphorylation, may be regulated by the associating active LTR5_Hs loci. The authors further speculated that LTR5_Hs insertions may have altered expression of their adjacent genes towards the pattern specific to the hominoid PGCs/PGCLCs, playing a unique evolutionary role in hominoid PGC development. Interestingly, our recent study revealed that HML-2 loci activated in human PGCLCs harbored LTR5_Hs in both 5’ and 3’ LTRs whereas HML-2 containing LTR5A or LTR5B at one or two viral LTRs remained silent [15], suggesting that the HML-2/LTR5_Hs loci may have unique significance in the evolutional aspects of PGC development whereas contributions of the older HML-2 loci common to hominoids and OWM seem limited.
In addition, we added the following sentence at the end of the concluding paragraph in Discussion.
Contributions of HML-2 in PGC(LC) differentiation may also provide insights into the evolutionary perspective of germline lineage specification in primates.
Reviewer 2 Report
Comments and Suggestions for Authors
This interesting manuscript presents data comparing the topologically associated domains within chromatin in primordial germ cell-like cells and induced pluripotent stem cells. The authors found rather subtle differences in sizes and distances of these domains; the degree of activation of HML-2 loci upon differentiation was inversely correlated with the distances between contacts. The data seem straightforward and of interest.
Author Response
We greatly appreciate the reviewer for his/her supportive comments and positive evaluations!
Reviewer 3 Report
Comments and Suggestions for Authors
Here, the authors differentiated human induced pluripotent stem cells (iPSCs) towards pluripotent stem cell-derived primordial germ cell-like cells (PGCLCs). Both cell populations were used for ligation-based Hi-C deep sequencing to identify topologically associated domains (TADs) and differential chromatin interactions (DCIs). The chromatin contacts were then associated with differentiation-upregulated members of the human endogenous retroviruses, here the HML-2 clade. The authors conclude that "the distances between chromatin contacts newly formed in PGCLCs and the degree of activation of the closest HML-2 loci showed significant inverse correlation." (Abstract) and "...our analysis shows that most strongly activated HML-2 loci are associated with newly formed TAD boundaries,..." (Introduction).
This study suffers from a number of limitations:
- only one clone of human iPSCs (A4) was used, and all the PGCLCs were derived from A4. The biological replicates are therefore equal to one, in order to generate meaningful data, three biological replicates are the minimum requirement.
-the clade of HML-2 consists of 21 members, thus less than one endogenous retrovirus per chromosome. Therefore it is unclear whether the correlation with the much more frequent DCIs and TADs makes any sense at all.
- the expression data of the HML-2 members seem to originate from a previous work of the authors (line 75), and it is unclear whether the identical cells were used for determining expression data and chromatin contacts.
In summary, the data seem to represent preliminary results, which must be substantiated by additional work.
Author Response
Comments 1 and 3: Only one clone of human iPSCs (A4) was used, and all the PGCLCs were derived from A4. The biological replicates are therefore equal to one, in order to generate meaningful data, three biological replicates are the minimum requirement.
The expression data of the HML-2 members seem to originate from a previous work of the authors (line 75), and it is unclear whether the identical cells were used for determining expression data and chromatin contacts.
Responses to Comments 1 and 3: We respond to comments 1 and 3 as they are closely relevant. We appreciate the reviewer for careful reading of the manuscript and pointing typographical errors. As the reviewer predicted, the HML-2 expression data presented in the current manuscript originate from our preceding work mentioned as Reference #15 (Kobayashi et al., bioRxiv). The lines 155-158 of the preprint (Ref#15) reads as follows:
“Using the ERVmap tool of quantitative determination of RNA expression from HERV loci (Tokuyama et al., 2018) and RNA-seq data we previously published (Matsunaga et al., 2017), we examined HERV RNA expression in the CD38-positive hPGCLCs, their precursor hiPSCs (clones A4, A5, A6), and CD38-negative non-germline cells.”
The human PGCLCs examined in the current study for chromatin contacts by Hi-C were indeed identical to clones mentioned in the preprint – namely, PGCLCs derived from human iPSC clones A4, A5, and A6. We addressed the review comments by amending several sentences in Results for clarity as follows:
“Human PGCLC cultures were generated from naïve iPSC clones in three independent experiments.”
" As shown in Figure 2A, the BART3D-identified DCIs were mostly persistent across the three biological experimental replicates.”
And then Materials and Methods as follows:
“Human PGCLCs were generated from the human iPSC clones A4, A5, and A6 using the protocol we previously described …. These three iPSC clones and human PGCLCs derived from them were identical to cells examined in our preceding study for expression of viral RNA from HERV loci [15].”
Please note that other parts of the Materials and Methods section and the Figure 2 legend already clarified that the three Hi-C data presented there are BIOLOGICAL replicates as follows (underlines added here for emphasis):
[Materials and Methods]
“Biological replicate PGCLC cultures were generated in three independent experiments performed at different times.”
“DCIs conserved across our three pairs of biological replicates (PGCLC vs. iPSC) were identified, …”
“The DCIs supported by both BART3D and ClusterTAD analyses of the three biological replicates were subjected to estimation of …”
[Figure 2 legend]
“Figure 2. (A, B) Circos plots illustrating the locations of conserved genomic regions and HML-2 sequences. (A) Conserved DCIs identified via BART3D across biological replicates. The three outermost rings depict DCIs with increased interactions in PGCLCs compared to naive iPSCs.”
“(B) Best TADs estimated with ClusterTAD that are closest to conserved DCIs. The outermost ring displays the conserved DCIs across replicate pairs. The next three rings show the best TADs from the three PGCLC biological replicates nearest to these conserved DCIs. The innermost ring (gray background) shows HML-2 locations.”
To these legend sentences, we added the following phrase for further clarification.
“(A4, A5, and A6 clones from the inner to outer rings).”
Comment 2: The clade of HML-2 consists of 21 members, thus less than one endogenous retrovirus per chromosome. Therefore it is unclear whether the correlation with the much more frequent DCIs and TADs makes any sense at all.
Response to Comment 2:
Our current study provides evidence that strongly activated HML-2 proviruses tend to locate in close vicinity of chromatin contacts that are newly formed upon differentiation of human iPSCs to PGCLCs. The number of the affected chromatin contacts can be very small, and they do not need to be found in each of all chromosomes. We appreciate the reviewer for pointing that our original manuscript was not sufficiently clear to convey the idea that HML-2 activation may contribute to a small fraction of DCIs or TADs newly formed or eliminated during differentiation of human iPSCs to PGCLCs. Many DCIs or differential TADs do not have apparent association with any HML-2 loci.
According to the study of Xue et al. entitled, “Identification of the distribution of human endogenous retroviruses K (HML‑2) by PCR‑based target enrichment sequencing. Retrovirology (2020),” the human genome contains at least 90 HML-2 provirus insertions (which may be capable of expressing viral RNA) and 1,015 solo-LTR insertions. Among them, 85 proviruses and 946 solo-LTRs were annotated in the human hg19 reference genome, which is an older version of the human reference genome sequence. Although the numbers of HML-2 proviruses and solo-LTR insertions show minor deviations among studies, roughly 100 and 1,000 HML-2 proviruses and solo-LTRs are commonly recognized. However, our preceding study (cited in the original manuscript as Ref# 15) revealed that only a fraction of the HML-2 provirus (~20 loci) is significantly activated in human PGCLCs. Activated HML-2 solo-LTR loci are not readily identified as they do not express viral RNA transcripts, which is admittedly a limitation of our current study.
To address this review comment, we added the following paragraphs in Discussion.
We identified ~120 DCIs and ~80 differential TADs between human PGCLCs and their precursor iPSCs (Figures 2A, 2B). Among the top 19 strongly activated HML-2 provirus loci identified in our previous study [15], the three most strongly expressed loci were located in very close vicinity of the differential TAD boundaries (1 – 7 Mbp; Expression Rank 1 in Figure 2C). More weakly expressed HML-2 provirus loci tend to be more distant from the differential TAD boundaries (Figure 2C). The association between strong HML-2 provirus expression and the presence of nearby differential TAD boundaries showed strong statistical significance: The Spearman’s correlation test (p = 0.0072) and a permutation test (p = 0.003) shown in Figures 2C and 2D, respectively]. Based on these observations, we speculate that strongly activate HML-2 proviruses may be capable of creating TAD boundaries. However, it is important to emphasize that the number of TAD boundaries whose creation may involve activated loci of HML-2 proviruses is only a fraction of the numbers of all TAD boundaries or DCIs. In our preceding study, we identified activated HML-2 provirus loci using RNA-seq followed by ERVmap2 locus-specific quantitation of viral RNA [15]. This approach was unable to detect potentially activated solo-LTR loci of HML-2 due to the lack of viral RNA expression. Because the number of solo-LTR HML-2 loci (~1,000) is far larger than that of RNA-expressing HML-2 proviruses (~100) [Xue et al. (2020) Retrovirology; new reference], the differential TADs or DCIs that were not linked to HML-2 provirus activation in the current study could still be associated with activation of solo-LTR HML-2 loci. Future study will be necessary to determine the complete links between the activated loci of HML-2 provirus and solo-LTR and the formation of TADs or DCIs upon differentiation of human PSCs to PGC(LC)s.
Round 2
Reviewer 3 Report
Comments and Suggestions for Authors
Most of the concerns from the first review were addressed in the revised version, however, the association between new established TADs and activated HML-2 retroviruses is still low and the authors should more clearly point out the limitations of their study.
Author Response
Comment: Most of the concerns from the first review were addressed in the revised version, however, the association between new established TADs and activated HML-2 retroviruses is still low and the authors should more clearly point out the limitations of their study.
Response: We agree with the reviewer. To address the comment, we revised the manuscript as follows:
[Abstract-Conclusion]
(Original) Our study provide evidence that strong activation of HML-2 may create new chromatin contact in their vicinity, potentially contributing to PSC differentiation to the germ cell lineage.
(Revised) Our study provide evidence that strong activation of HML-2 provirus loci may be associated with newly formed chromatin contacts in their vicinity, potentially contributing to PSC differentiation to the germ cell lineage.
[Discussion – To clarify limitations of the current study, we revised the relevant paragraph as follows.]
We identified ~120 DCIs and ~80 differential TADs between human PGCLCs and their precursor iPSCs (Figures 2A, 2B). Among the top 19 strongly activated HML-2 provirus loci identified in our previous study [15], the three most strongly expressed loci were located in very statistically significant close vicinity of the differential TAD boundaries (1 – 7 Mbp; Expression Rank 1 in Figure 2C). More weakly expressed HML-2 provirus loci tend to be more distant from the differential TAD boundaries (Figure 2C). The inverse association between strong the strength of RNA expression from the HML-2 provirus expression loci and the presence of nearby differential newly formed TAD boundaries showed strong statistical significance was also statistically significant: The Spearman’s correlation test (p = 0.0072) and a permutation test (p = 0.003) shown in Figures 2C and 2D, respectively]. Based on these observations, we speculate that strongly activatestrong activation of HML-2 proviruses may be capable of creating contribute to creation of new TAD boundaries in PGCLCs. However, because the observed association is relatively low, our current study could not determine whether HML-2 activation directly contributes to formation of new TAD boundaries. It is possible that strong activation of HML-2 opens otherwise tightly closed chromatin, thus increasing the chance of TAD formation in the relatively close vicinity but via different mechanisms that may not be directly tied with the HML proviruses. On the other hand, it is also possible that formation of new TAD boundaries via different mechanisms reduces silencing of HML-2 proviruses by heterochromatin. Future studies will be necessary to clarify the mechanisms behind the apparent association between HML-2 provirus activation and formation of new TAD boundaries. However, it is important to emphasize that the The number of TAD boundaries whose creation may involve be associated with the activated loci of HML-2 proviruses is only a fraction of the numbers of all TAD boundaries or DCIs. In our preceding study, we identified activated HML-2 provirus loci using RNA-seq followed by ERVmap2 locus-specific quantitation of viral RNA [15]. This approach was unable to detect potentially activated solo-LTR loci of HML-2 due to the lack of viral RNA expression. Because the number of solo-LTR HML-2 loci (~1,000) is far larger than that of RNA-expressing HML-2 proviruses (~100) [Xue et al. (2020) Retrovirology; new reference], the differential TADs or DCIs that were not linked to HML-2 provirus activation in the current study could still be associated with activation of solo-LTR HML-2 loci. Future study will be necessary to determine the complete links between the activated loci of HML-2 provirus and solo-LTR and the formation of TADs or DCIs upon differentiation of human PSCs to PGC(LC)s.
Round 3
Reviewer 3 Report
Comments and Suggestions for Authors
All points have been addressed.